# Effects of Vitamin D_3_ in Long-Term Ovariectomized Rats Subjected to Chronic Unpredictable Mild Stress: BDNF, NT-3, and NT-4 Implications

**DOI:** 10.3390/nu11081726

**Published:** 2019-07-26

**Authors:** Alexandra Koshkina, Tatyana Dudnichenko, Denis Baranenko, Julia Fedotova, Filippo Drago

**Affiliations:** 1Faculty of Food and Bioengineering, ITMO University, 49 Kronverksky Pr., St. Petersburg 197101, Russia; 2Department of Obstetrics and Gynecology, North-Western State Medical University, 41 Kirochnaya Str., St. Petersburg 191015, Russia; 3Laboratory of Neuroendocrinology, I.P. Pavlov Institute of Physiology Russian Academy of Sciences, 6 Emb. Makarova, St. Petersburg 199034, Russia; 4Department of Biomedical and Biotechnological Sciences, Biological Tower, School of Medicine, University of Catania, Via S. Sofia 97, Catania 95123, Italy

**Keywords:** vitamin D_3_, long-term ovariectomy, chronic unpredictable mild stress, BDNF, NT-3, NT-4

## Abstract

The purpose of this study was to explore the antidepressant-like effects of vitamin D_3_ at different doses (1.0, 2.5, and 5.0 mg/kg sc) on a model of depression produced by chronic unpredictable mild stress (CUMS) for 28 days in long-term (3 months) ovariectomized (OVX) adult rats. Sucrose preference (SPT), forced swimming (FST) and open-field (OFT) tests were conducted to examine the depression-like state. Serum corticosterone/adrenocorticotrophic hormone (ACTH) levels and hippocampal brain-derived neurotrophic factor (BDNF) and neurotrophin (NT)-3/NT-4 expressions by ELISA kits and/or western blotting were determined to assess the possible mechanisms of the vitamin D_3_ effects on the depression-like profile in long-term OVX rats subjected to CUMS. The results showed that vitamin D_3_ (5.0 mg/kg), as well as fluoxetine treatment, considerably reversed the depression-like state in the SPT and FST, decreased serum corticosterone/ACTH levels, and increased BDNF and NT-3/NT-4 levels in the hippocampus of long-term OVX rats compared to OVX rats with CUMS (*p* < 0.05). Thus, a high dose of vitamin D_3_ (5.0 mg/kg sc) could improve the depression-like profile in long-term OVX adult female rats subjected to the CUMS procedure, which might be mediated by the regulation of BDNF and the NT-3/NT-4 signaling pathways in the hippocampus, as well as the corticosterone/ACTH levels of the blood serum.

## 1. Introduction

The menopausal period—a very specific state for women—is produced by the decreased secretion of the female gonadal hormones by the ovaries [1,2,3]. Numerous data of preclinical and clinical studies have shown that estrogen deficiency during menopause increases the susceptibility to mood disturbances, including depression [4,5,6,7]. On the other hand, the menopausal state in ovariectomized (OVX) female rodents and menopausal women is also characterized by vitamin D (VD) deficiency or insufficiency [8,9,10,11]. Our previous experimental work has confirmed that OVX rats after a long-term ovariectomy are characterized by lower serum 25-OH-VD levels [9,12].

It is generally accepted that VD is an essential substance needed for the homeostasis of calcium and phosphorus in the human body [13,14]. Nowadays, VD is postulated as one of the neurosteroids that may be implicated in the development of various neurodegenerative and affective-related disorders [15]. The functionally active form of VD (1.25-OH-VD_2_) reveals both genomic and nongenomic effects [13,14,16]. VD receptors (VDR) and VD-activating enzymes have been found in different brain structures, including the hippocampus, prefrontal cortex, and amygdala [17,18]. Since VDR and the 1-alpha-hydroxylase enzyme that transforms 25-OH-VD to 1.25 (OH) VD_2_ are identified in the limbic system, this fact gives some explanations of the relationship between VD levels and mood disturbances [17]. Moreover, VD might modulate different physiological processes in the brain, such as the regulation of brain-derived neurotrophic factor (BDNF) and other neurotrophic factors, neurogenesis, neuroplasticity, neuroprotection, and neuroimmunomodulation [19,20], thereby involving the pathophysiological mechanisms of depressive disorders [21,22,23,24,25]. Several observations suggest that serum 25-VD levels are considered as prognostic markers for increased morbidity and mortality [26]. However, some studies have demonstrated obvious results between VD status and depressive symptoms [27].

One of the key structures of the brain that is involved in affective-related disorders is the hippocampus [28]. Furthermore, the hippocampus has been assumed as a target of both estrogen and VD_3_ [25,26]. The alteration of neurotrophic factors and their expression in the hippocampus are associated with depression in both human and animal models [29,30,31]. Preclinical studies in animals have indicated that the depression-like profile in OVX rats is correlated with decreased BDNF levels in the limbic system of the brain [32,33]. Clinical studies using depressed patients have shown that brain-derived neurotrophic factor (BDNF) levels are diminished in the serum of such patients [34,35,36,37,38].

On the other hand, it is well known that hypothalamic-pituitary-adrenal (HPA) axis hyperactivity is supposed to be one of the major trigger factors for the development of mood disorders [39,40,41]. Taking this assumption into account, depression established in menopausal women might result from multitarget alterations in HPA activity, estrogens, and VD_3_ levels, as well as BDNF and other neurotrophin production. Our previous studies confirmed that VD_3_ at doses of 1.0 and 2.5 mg/kg sc alone, or with the addition of a low dose of 17β-estradiol, shortened the immobility time during a forced swimming test in adult OVX rats at 2 or 12 weeks after surgery [42], suggesting that VD_3_ exhibits an antidepressant-like effect on rat behavioral despair depression models. However, the therapeutic effects of VD_3_ on the chronic unpredictable mild stress (CUMS) model of depression, and whether the antidepressant-like action of VD_3_ involves BDNF, neurotrophin-3 (NT-3), neurotrophin-4 (NT-4), and the HPA axis in long-term OVX adult rats remain unknown.

Therefore, the purpose of this study was to explore the antidepressant-like effect of VD_3_ at different doses by the implications of the BDNF and neurotrophin signaling pathways, as well as the HPA axis, on a rat model of depression produced by CUMS. In this study, we used long-term estrogen deficiency caused by a post-ovariectomy period of 3 months, similar to which has been published previously [43]. This animal model is widely utilized in preclinical behavioral research producing a menopausal-like state in women [44]. Sucrose solution consumption tests, as well as forced swimming and open-field tests, were conducted to examine the depression-like state. Serum corticosterone (CS) and adrenocorticotrophic hormone (ACTH) levels and hippocampal BDNF, NT-3, and NT-4 concentrations were determined to assess the possible mechanisms of the VD_3_ effects on the depression-like profile in long-term OVX rats subjected to CUMS.

## 2. Experimental Section

### 2.1. Animals

Adult female Wistar rats (weighing 210 ± 20 g) were purchased from the Animal Rat Center of the Rappolovo Laboratory Animal Factory (St. Petersburg, Russia). All females were maintained under standard animal vivarium conditions with a constant room temperature (22 ± 1 °C), relative humidity (50 ± 10%), and a 12 h light/dark cycle (light from 07:00 to 19:00) with typical food for rodents and water ad libitum. All rats were allowed to habituate to the novel environment for 1 week prior to their use in experiments. The experiments were carried out in compliance with the National Institute of Health guidelines for laboratory animals and were approved by the Animal Care Committee of the I.P. Pavlov Institute of Physiology (Protocol No.: 1095/1/25.06.2012). All stress manipulations were performed to minimize any pain and undesirable experiences in the experimental animals.

### 2.2. Ovariectomy

In the present study, the long-term bilateral removal of ovaries in female rats was used to modulate the hormonal state similar to the menopausal period in women [44,45]. A narcosis was performed by the intraperitoneal administration of ketamine 70 mg/kg and xylazine 10 mg/kg. The operation was performed as described previously [43]. Both ovaries were removed using two standard cuts in a lateral position. Then, the muscles and skin incision were restored by surgical staples. For the sham operation, all manipulations were repeated, however, without the amputation of the ovaries. The efficacy of the surgery was validated by routine vaginal inspection and serum estradiol levels. Following the ovariectomy or sham operation, the OVX females were placed in a home cage with free access to food and water. After the ovariectomy, the recovery period was continued for 12 weeks. Following this time, each rat was randomly assigned to an experimental group for the chronic stress procedure, except for the sham-operated (SHAM), non-stressed, control rats.

### 2.3. CUMS Model

Chronic unpredictable mild stress (CUMS) is a significant model of clinical depression in laboratory animals [46]. CUMS was created as described previously with some alterations [47,48]. The procedure included exposure to different and unpredictable stressors that were randomly changed during the experiment [49]. The total scheme of the experiment is indicated in Figure 1.

These manipulations were 24 h food deprivation, 24 h water deprivation, wet bedding overnight, tilted cage overnight, unpredictable shocks (15 mA, one shock/20 s, 10 s duration, 20 min), 5 min swimming in cold water (4 °C), tail hanging for 1 min, clipped tail for 1 min, and reversal of light/dark cycle [47,48]. All the stress triggers were performed individually and continuously. To prevent habituation and to ensure the unpredictability of the stressors, all stress manipulations were randomly made according to the experimental scheme and repeated throughout the four weeks of the CUMS protocol. The control, sham-operated females were placed in a separate room without any contact with the stressed groups of animals. These rats were maintained as undisturbed animals and were subjected only to routine cage cleaning for 4 weeks. The rats were weighed both before and after the CUMS period.

### 2.4. Drugs

Fluoxetine hydrochloride and VD_3_, as cholecalciferol, were provided by Sigma Chemical Co. (St. Louis, MO, USA). VD_3_ was dissolved in 95% ethanol, then aliquoted, and remained at –80 °C. The solution of cholecalciferol for the injection into the experimental groups was diluted in sterile water, resulting in a solvent of VD_3_ containing 2% ethanol. Fluoxetine hydrochloride was dissolved in sterile physiological saline. All drugs were injected subcutaneously (0.1 mL/rat) for the 4 weeks during the CUMS procedure—30 min before the daily stressor action—and throughout the period of the behavioral tests. All behavioral measurements were made 60 min after the last drug administration.

### 2.5. Groups of Animals

The animals were randomly assigned to the experimental groups (*n* = 7 in each): SHAM rats without the CUMS model treated with saline (control), SHAM rats submitted to CUMS treated with saline, long-term OVX rats exposed to CUMS given with saline, fluoxetine as positive control (10.0 mg/kg/day) or VD_3_ (1.0, 2.5, 5.0 mg/kg/day). In our preliminary studies, there were no significant differences between SHAM/OVX rats treated with physiological saline as a solvent for fluoxetine and SHAM/OVX females treated with sterile water with 2% ethanol as a solvent for VD_3_ in behavioral trials (data are not shown). Since, we did not found any differences between these experimental groups, physiological saline as a solvent for SHAM/OVX females was used in the present work. The doses of VD_3_ were based on our previous studies on the behavioral effects of VD_3_ on depression-like behavior of non-stressed long-term OVX female rats [42]. The dose of fluoxetine was utilized according to earlier experimental data [50]. Several studies have demonstrated that the administration of fluoxetine decreases depressive-like behavior in rodents [50,51]. All drugs were injected subcutaneously (0.1 mL/rat) for the 4 weeks during the CUMS procedure—30 min before the daily stressor action—and throughout the period of the behavioral tests. All behavioral measurements were made 60 min after the last drug administration.

### 2.6. Sucrose Preference Test

Before the initiation of CUMS and after 4 weeks of stress procedures, the experimental rats were examined by the sucrose preference test (SPT) [52,53,54]. Following a training trial, the rats were subjected to a deprivation of food and water for 24 h. On the next day, the rats had free access to one bottle with 200 mL of sucrose solution and another bottle with a similar volume of water. One hour later, the parameters of the consumed sucrose solution and water volumes were registered. The value of the sucrose preference in percentage was calculated as the amount of sucrose solution consumed (mL) among all (sucrose plus water in mL) liquid consumption:$$\%\;\mathrm{sucrose}\;\mathrm{preference}=\frac{\mathrm{sucrose}\;\mathrm{consumption}}{\mathrm{sucrose}\;\mathrm{consumption}\;+\;\mathrm{water}\;\mathrm{consumption}}\times 100$$

### 2.7. Forced Swimming Test

To test the modifications of depression-like behavior, OVX female rats with CUMS were submitted to the standard forced swimming test (FST) as described in earlier work [42]. Three cylinders (60 cm tall and with a diameter of 20 cm) were filled with water at 23–25 °C up to a 30 cm depth. On the first day, rats were pre-tested for 15 min in the cylinders. Then, the rats were dried with paper and placed in their home cages until the next day. On the second day (testing trial), the OVX females with CUMS were examined in the apparatus for 5 min. The following parameters were registered: (1) immobility time (floating in the water with only movements necessary to keep the head above water), (2) swimming time (active swimming movements around the glass cylinder), and (3) climbing time (active movements with forepaws directed toward the walls). To record these values, a video camera was installed above the apparatus.

### 2.8. Open Field Test

The measurements of the behavioral activity in the OFT were carried out in a similar way to the method which has been published in a previous study [43]. The rats were set in the center square of the OFT and tested for 5 min. Motor activity and rearing and grooming behavior were recorded for 300 s in the OFT apparatus using a video camera. After each rat had been in the OFT, the apparatus was de-contaminated using a cleaning solution.

### 2.9. ELISA Measurements

After all the behavioral testing, approximately 5 mL samples of blood were drawn from the animals decapitated by a similar narcosis as used for the ovariectomy. Simultaneously, the hippocampi of each experimental group of rats were immediately dissected on dry ice. Blood samples were centrifuged at 4000 *g* for 15 min at 4 °C. The hippocampi of each experimental group were homogenized in cold lysis extraction buffer (0.2% sodium deoxycholate, 0.5% Triton X-100, 1% NP-40, 50 mM Tris–HCl pH 7.4, 1 mM phenylmethylsulfonyl fluoride, 1 mM N-ethyl-maleimide, and 2.5 mM phenantroline) [55]. After that, the hippocampal samples with the cold lysis buffer were sonicated for 15 s. Then, the hippocampi were centrifuged at 12,000 *g* for 15 min at 4 °C. The Bradford method was used for the normalization of hippocampal supernatants to the total protein [56]. The serum samples and hippocampal protein normalized supernatants were stored at −80 °C until the ELISA assays. The serum samples were used for the measurement of the 25-hydroxyvitamin D_3_ (25-OH-VD_3_), ACTH, corticosterone, and estradiol levels using a commercially available rat ELISA kits (Cusabio Biotech Co., Ltd., Wuhan, China) according to the manufacturer’s instructions. The sensitivity and detection range of the 25-OH-VD_3_ rat ELISA kits were 5.0 µg/L and 20–100 µg/L, respectively. The sensitivity and detection range of the corticosterone rat ELISA kits were 0.1 ng/mL and 0.2–40 ng/mL, respectively. The sensitivity and detection range of the ACTH rat ELISA kits were 1.25 pg/mL and 1.25–50 pg/mL, respectively. The sensitivity and detection range of the estradiol rat ELISA kits were 4.0 pg/mL and 40–1500 pg/mL, respectively.

Hippocampal homogenates were used for the detection of the BDNF, NT-3, and NT-4 levels by rat ELISA kits (Cusabio Biotech Co., Ltd., Wuhan, China) according to the manufacturer’s instructions. Briefly, 100 μL of hippocampal sample or standard was added to each well and incubated for 120 min at 37.0 °C. Then, 100 μL of anti-BNDF, anti-NT-3, or anti-NT-4 antibodies were added to each different well and incubated for 60 min at 37.0 °C. After three times of washing, 100 μL of HRP–avidin working solution was added to each well and incubated for 60 min at 37.0 °C. Again, after five times of washing, 90 μL of tetramethylbenzidine solution was given to each different well and incubated for 15–30 min at 37.0 °C. Then, 50 μL of stop solution was added to each well to terminate the color reaction. The BDNF, NT-3, or NT-4 levels were measured using an MC Thermo Fisher Scientific reader (Thermo Fisher Scientific Inc., Helsinki, Finland) with an absorbance of 450 nm. The standard curve was used for the calculation of the relationship between the optical density and the BDNF and NT-3/NT-4 levels. The BDNF and NT-3/NT-4 contents are presented as pg/mg of tissue. The sensitivity and detection range of the BDNF rat ELISA kits were 0.078 ng/mL and 0.312–20 ng/mL, respectively. The sensitivity and detection range of the NT-3 rat ELISA kits were 0.225 ng/mL and 0.9–60 ng/mL, respectively. The sensitivity and detection range of the NT-4 rat ELISA kits were 11.75 pg/mL and 47–3000 pg/mL, respectively. The assay exhibited no significant cross-reactivity with other neurotrophic factors. All samples were duplicated for the assay.

### 2.10. Western Blotting Analysis

Hippocampal tissues were homogenized in cold lysis buffer containing a protease inhibitor cocktail (Sigma-Aldrich, St. Louis, MO, USA) for 1 h and centrifuged at 12,000 *g* at 4 °C for 20 min. The protein content was evaluated by a Bio-Rad protein detector (Bio-Rad, Hercules, CA, USA), and 100 µg of total protein from each sample was denatured with buffer (6.205 mM Tris–HCl, 10% glycerol, 2% SDS, 0.01% bromophenol blue, and 50 mM 2 ME) at 95 °C for 5 min. The denatured proteins were separated on an SDS page (10% sodium dodecyl sulfate-polyacrylamide gel) and forwarded to a nitrocellulose membrane (Amersham Biotech, Little Chalfont, UK). After that, the membranes were probed with anti-BDNF, anti-NT4, anti-NT-3 (1:1000, Santa Cruz, CA, USA), and β-actin (1:1000; Sigma-Aldrich, St. Louis, MO, USA) monoclonal antibodies for 2 h and secondary antirabbit antibodies (1:5000; Santa Cruz, CA, USA) conjugated to horseradish peroxidase (for BDNF and NT-3/NT-4) for 1 h. Bands were detected by 5-bromo-4-chloro-3-indolyl phosphate with a nitro blue tetrazolium kit (Abcam, Shanghai, China) as a chemiluminescent substrate. Signals were measured by an image analysis system (UVIdoc, Houston, TX, USA).

### 2.11. Statistical Analysis

All experimental data are expressed as the mean ± standard deviation of the mean. The treatment effects were determined with a one-way ANOVA followed by an LSD post hoc test using the Statistics Package for SPSS, version 16.0 (SPSS Inc., Chicago, IL, USA). A value of *p* < 0.05 was considered statistically significant.

## 3. Results

### 3.1. The Effects of VD_3_ Administered in Various Doses on the Body Weight in the Long-Term OVX Rats Subjected to CUMS

The body weights of long-term OVX rats subjected to CUMS and treated with different doses of VD_3_ are indicated in Figure 2. There was no difference in the initial body weight in all the experimental groups. Following 4 weeks, the bodyweight of SHAM rats with CUMS was significantly decreased compared to the control, non-CUMS SHAM group (Figure 2, F (1,76) = 65.43, *p* < 0.001). The bodyweight of long-term OVX rats with CUMS was statistically reduced compared to the non-CUMS/CUMS SHAM groups (Figure 2, *p* < 0.001). The high VD_3_ administration (5.0 mg/kg) significantly prevented the reduction of the body weight (*p* < 0.001) and was similar to the effect of the positive drug fluoxetine (10.0 mg/kg). However, the low dose of VD_3_ (1.0 mg/kg) induced a more marked reduction of body weight in the long-term OVX rats compared to the OVX/SHAM rats with CUMS group (Figure 2, *p* < 0.001). The middle dose (2.5 mg/kg) of VD_3_ was not effective in the restoration of body weight in the long-term OVX rats with CUMS (Figure 2, F (1,76) = 0.26, *p* > 0.05), which is consistent with the results in the SPT, FST, and OFT.

### 3.2. Effects of VD_3_ Administered in Various Doses on Sucrose Preference in the Long-Term OVX Rats Subjected to CUMS

The SPT is frequently utilized to evaluate the hedonic state in rodents. There was no significant difference in the sucrose preference among groups prior to the CUMS procedure (Figure 3). After 4 weeks of CUMS, the SHAM rats demonstrated the reduction in sucrose preference compared to the non-CUMS control SHAM group (*p* < 0.01).

The OVX rats showed a significant decrease in sucrose preference compared to the non-CUMS/CUMS SHAM rats (Figure 3, F (1,76) = 42.75, *p* < 0.01). VD_3_ (5.0 mg/kg), as well as fluoxetine treatment, considerably enhanced the consumption of sucrose in the long-term OVX rats submitted to CUMS compared to the OVX/SHAM rats with CUMS (Figure 2, *p* < 0.01). Interestingly, the sucrose preference of the long-term OVX with CUMS plus VD_3_ (1.0 mg/kg) was lower than that of the OVX/SHAM with CUMS and non-CUMS control SHAM groups (Figure 3, *p* < 0.01). This result demonstrated severe depression in the long-term OVX females with CUMS plus VD_3_ at a dose of 1.0 mg/kg. Moreover, there was no statistical difference between the long-term OVX/SHAM with CUMS plus saline and the OVX with CUMS plus VD_3_ (2.5 mg/kg) groups (Figure 3, F (1,76) = 1.13, *p* > 0.01).

### 3.3. Effects of VD_3_ Administered in Various Doses on Depression-Like Behavior in the Forced Swimming Test of Long-Term OVX Rats Subjected to CUMS

CUMS produced a significant increase of the immobility time and decrease of swimming time in the long-term OVX compared to the non-CUMS/CUMS SHAM rats (Figure 4, F (1,76) = 57.81, F (1,76) = 67.89, F (1,76) = 22.17, respectively, *p* < 0.001). VD_3_ (5.0 mg/kg), as well as fluoxetine treatment, significantly reduced the immobility time and increased the swimming time in the long-term OVX compared to the OVX/SHAM with CUMS groups (Figure 4, *p* < 0.001). However, VD_3_ (1.0 mg/kg) exacerbated depression-like behavior in the long-term OVX rats subjected to CUMS compared to the long-term OVX/SHAM rats subjected to CUMS and the non-CUMS SHAM, control animals (Figure 4, *p* < 0.001). We did not find any effects of VD_3_ (2.5 mg/kg) administration on the depression-like parameters of the long-term OVX rats exposed to CUMS in the FST compared to the long-term OVX with CUMS group plus saline (Figure 4, *p* > 0.05). There was no difference in the climbing time in all the experimental groups compared to the OVX/SHAM with CUMS groups (Figure 4, *p* > 0.05).

### 3.4. Effects of VD_3_ Administered in Various Doses on Behavior in the open Field Test of Long-Term OVX Rats Subjected to CUMS

Following 4 weeks of CUMS, there were no statistically significant differences for grooming activities between all the experimental groups of animals in the OFT (Figure 5, F (1,76) = 0.78, *p* > 0.05). 

The long-term OVX rats with CUMS showed a reduced number of rearings and crossings when they were compared to the non-CUMS/CUMS SHAM groups (Figure 5, F (1,76) = 11.12, *p* < 0.05). Administration of fluoxetine, as well as treatment with VD_3_ at various doses, significantly increased the number of rearings and crossings in the long-term OVX with CUMS compared to the OVX/SHAM with CUMS plus saline groups (Figure 5, *p* < 0.05).

### 3.5. Effects of VD_3_ Administered in Various Doses on Serum Corticosterone, ACTH, VD_3_ and Estradiol Levels in Long-Term OVX Rats Subjected to CUMS

The ELISA assays revealed that serum CS and ACTH levels were elevated, as well as decreased estradiol and VD_3_ levels in the long-term OVX rats with CUMS compared to the non-CUMS/CUMS SHAM groups (Figure 6, F (1,76) = 122.74, F (1,76) = 34.46, F (1,76) = 14.45, and F (1,76) = 13.60, respectively, *p* < 0.001). The administration of VD_3_ (1.0, 2.5, and 5.0 mg/kg) dose-dependently reversed the pathologically enhanced CS and ACTH levels in the blood serum of the long-term OVX rats exposed to CUMS compared to the OVX/SHAM rats subjected to CUMS plus saline (Figure 6, *p* < 0.001). However, the CS/ACTH concentrations in all groups of the long-term OVX rats after CUMS administered with VD_3_ or fluoxetine were not able to decrease to the values of non-CUMS control rats. Treatment with VD_3_ dose-dependently increased serum VD_3_ levels in the long-term OVX rats with CUMS compared to the OVX group exposed to CUMS plus saline (Figure 6, *p* < 0.001). Fluoxetine did not change the serum VD_3_ level but significantly reduced the serum CS and ACTH levels in the long-term OVX rats exposed to CUMS compared to the OVX/SHAM groups with CUMS plus saline (Figure 6, *p* < 0.001). A one-way ANOVA test failed to demonstrate that serum estradiol levels of long-term OVX rats exposed to CUMS were modified by VD_3_ or fluoxetine compared to the OVX rats with CUMS plus saline (Figure 6, F (1,76) = 0.78, *p* > 0.05).

### 3.6. Effects of VD_3_ Administered in Various Doses on Hippocampal BDNF, NT-3, NT-4 Levels in Long-Term OVX Rats Subjected to CUMS

CUMS reduced BDNF concentration in the hippocampus of SHAM rats compared to the non-CUMS control females (Figure 7, *p* < 0.001). CUMS produced a decrease of hippocampal BDNF and NT-3/NT-4 levels in the long-term OVX rats compared to the non-CUMS/CUMS SHAM rats (Figure 7, F (1,76) = 49.12, F (1,76) = 17.11, F (1,76) = 9.56, respectively, *p* < 0.05). Supplementation with VD_3_ (5.0 mg/kg) or fluoxetine (10.0 mg/kg) increased BDNF in the hippocampus of long-term OVX rats exposed to CUMS compared to the OVX/SHAM rats with CUMS (Figure 7, *p* < 0.05). Moreover, both VD_3_ and fluoxetine elevated NT-3/NT-4 concentrations in the hippocampus of the long-term OVX females exposed to CUMS compared to the OVX rats with CUMS plus saline (Figure 7, *p* < 0.05). The administration of VD_3_ (1.0 mg/kg) induced much lower BDNF content in the hippocampus of the long-term OVX with CUMS group compared to the OVX/SHAM with CUMS plus saline and non-CUMS SHAM plus saline groups (Figure 7, *p* < 0.05). Furthermore, VD_3_ (1.0 mg/kg) failed to alter NT-3/NT-4 levels in the hippocampus of the long-term OVX rats with CUMS compared to the OVX rats with CUMS plus saline (Figure 7, *p* > 0.05). Application with VD_3_ (2.5 mg/kg) did not modify BDNF, NT-3, and NT-4 levels in the hippocampus of the long-term OVX rats with CUMS compared to the OVX rats with CUMS plus saline (Figure 7, *p* > 0.05).

### 3.7. Effects of VD_3_ Administered in Various Doses on Protein Expressions of BDNF, NT-3, and NT-4 in Long-Term OVX Rats Subjected to CUMS

Findings from western blotting determination showed that CUMS diminished BDNF protein levels in the hippocampus of SHAM rats compared to non-CUMS control females (Figure 8, *p* < 0.001). BDNF and NT-3/NT-4 protein levels were decreased in the hippocampus of long-term OVX rats with CUMS compared to the non-CUMS/CUMS SHAM rats (Figure 8, F (1,76) = 14.84, F (1,76) = 11.29, F (1,76) = 8.87, respectively, *p* < 0.01). VD_3_ (5.0 mg/kg) or fluoxetine (10.0 mg/kg) resulted in significant elevated levels of hippocampal BDNF in long-term OVX rats compared to the OVX/SHAM rats with CUMS (Figure 8, *p* < 0.01). Moreover, application with VD_3_ or fluoxetine elevated NT-3/NT-4 protein levels expressed in the hippocampus of OVX rats submitted to CUMS compared to the OVX rats with CUMS plus saline (Figure 8, *p* < 0.01). Neither VD_3_ (5.0 mg/kg) nor fluoxetine changed the NT-3/NT-4 protein expression levels in the hippocampus of the long-term OVX rats compared to the non-CUMS/CUMS SHAM rats (Figure 8, *p* > 0.01). The long-term OVX with CUMS plus VD_3_ (1.0 mg/kg) group displayed more significantly reduced BDNF and NT-3/NT-4 protein levels in the hippocampus compared to the OVX with CUMS plus saline and/or non-CUMS/CUMS SHAM groups (Figure 8, *p* < 0.01). The supplementation of VD_3_ (2.5 mg/kg) did not change the BDNF and NT-3/NT-4 protein expressions in the hippocampus of long-term OVX rats compared to the OVX with CUMS plus saline (Figure 8, *p* > 0.01).

## 4. Discussion

The present study was designed to explore the antidepressant-like effects of VD_3_ administered at different doses (1.0, 2.5, and 5.0 mg/kg sc) in long-term adult OVX rats exposed to CUMS. Moreover, we evaluated the involvement of BDNF and NT-3/NT-4 signaling pathways, as well as the HPA axis, in the mechanisms of VD_3_ action on the behavioral state of long-term adult OVX rats exposed to CUMS. A CUMS model is a standard animal model of stress-induced depression that is represented by forecasted validity [47,48]. The CUMS model has been demonstrated to reflect a typical pathological deterioration relevant to depression in clinics [49].

The findings of this study showed that in the adult long-term OVX rats undergoing CUMS there were marked anhedonia-like and depression-like behaviors, as evaluated by SPT and FST, respectively. Moreover, long-term OVX rats exposed to CUMS exhibited decreased locomotor and rearing activities in the OFT. The ELISA assay clearly demonstrated elevated serum CS/ACTH levels, as well as lower VD_3_ concentrations, in adult long-term OVX rats subjected to CUMS. In addition, the decreased BDNF and NT-3/NT-4 concentrations and BDNF and NT-3/NT-4 protein expressions were revealed using rat ELISA kits or western blotting in the hippocampus of long-term OVX rats exposed to CUMS. The results of the study confirm that CUMS produces marked behavioral, neuroendocrine, and neuroplasticity changes in adult OVX rats with long-term ovarian hormone deficiency (post-ovariectomy period of 3 months). Our data are in agreement with another finding, which indicated that long-term estrogen deprivation in female rodents subjected to a CUMS procedure results in a profound depressive-like profile [57].

Treatment with a positive reference drug (fluoxetine) decreased anhedonia-like and depression-like states and reversed neuroendocrine impairments in the long-term OVX female rats exposed to CUMS. Preclinical findings have documented that fluoxetine might restore the functional activity of the HPA axis and BDNF expression in different structures of the brain, improving the depression-like profile of OVX rats with different post-ovariectomy intervals in stressed and non-stressed models of depression [58].

The most important results of the present study are associated with the antidepressant-like effects of VD_3_ at several doses on the model of depression produced by CUMS in the long-term adult OVX rats. To our knowledge, our study is the first comparative investigation of VD_3_ action on the behavioral and neurochemical consequences of a CUMS procedure in adult long-term OVX rats.

VD_3_ administered at a dose of 5.0 mg/kg reversed anhedonia-like and depression-like states in the SPT/FST paradigms in the long-term OVX rats subjected to CUMS, which was similar to the effects of the fluoxetine treatment. Moreover, the VD_3_ application (5.0 mg/kg sc) restored the behavioral impairments observed in the OFT in the long-term OVX rats subjected to CUMS. Biochemical assays found that VD_3_ at this dose decreased the serum CS/ACTH concentrations, increased the serum VD_3_ and the hippocampal BDNF and NT-3/NT-4 levels in the long-term OVX rats exposed to CUMS. Western blotting revealed that VD_3_ (5.0 mg/kg sc) enhanced the hippocampal BDNF protein expression in the long-term OVX rats with CUMS. These data suggest that VD_3_ at a dose of 5.0 mg/kg sc attenuates the CUMS-induced behavioral impairments, improved the hormonal state, and restored the serum VD_3_ and neurotrophic factor levels in the hippocampus of long-term OVX rats.

However, the behavioral and neuroendocrine effects of vitamin D_3_ at doses of 1.0 or 2.5 mg/kg were strongly different from the effects of VD_3_ at a dose of 5.0 mg/kg. VD_3_ at a dose of 2.5 mg/kg failed to alter the behavioral and neuroendocrine characteristics of the long-term OVX rats exposed to CUMS. In contrast, VD_3_ supplementation at a dose of 1.0 mg/kg exacerbated the behavioral disturbances, inducing more pronounced anhedonia-like and depression-like profiles in the long-term OVX rats with CUMS. The fact that the VD_3_ application in all tested doses increased the rearings and crossings allows us to make a conclusion that the effects of VD_3_ in the SPT and FST cannot be attributed to behavioral changes in the OFT but can be supposed as direct actions on the anhedonia-like and depression-like profiles of the long-term OVX rats exposed to CUMS. Furthermore, the VD_3_ treatment (1.0 mg/kg sc) significantly reduced BDNF concentrations, as well as the protein expressions of all neurotrophic factors in the hippocampus of the long-term OVX rats with CUMS. Although, VD_3_ (1.0 mg/kg sc) markedly decreased CS/ACTH and increased VD_3_ levels in the blood serum of the long-term OVX rats exposed to CUMS, similar to the action of VD_3_ at a dose of 5.0 mg/kg, we were not able to register any improvements of the depression-like state in these rats. These results suggest that the effects of VD_3_ at doses of 1.0 and 5.0 mg/kg on the behavioral consequences of a CUMS procedure might involve its actions on the BDNF and NT-3/NT-4 signaling pathways in the hippocampus rather than the action of VD_3_ at these doses on the HPA axis or VD_3_ levels in long-term OVX rats.

In summary, we found an inversed dependence between the behavioral effects and doses of VD_3_—the dose at 1.0 mg/kg exacerbated anhedonia-like and depression-like profiles, 2.5 mg/kg was not effective, and 5.0 mg/kg reversed all the behavioral impairments in the long-term OVX rats exposed to CUMS. An adaption process may contribute to a better response for VD_3_ administration only at a dose of 5.0 mg/kg in adult long-term OVX female rats with CUMS. However, our earlier study demonstrated that VD_3_ treatment only at a dose of 1.0 mg/kg induced a decrease of depression-like behavior, while a dose of 2.5 mg/kg induced a pro-depressant-like effect and a dose of 5.0 mg/kg was not effective in the long-term, non-stressed OVX rats [42]. Thus, the data of the present study when using long-term OVX rats subjected to CUMS are completely opposite to the findings of our previous study concerning the behavioral effects of VD_3_ at similar doses in non-stressed long-term OVX rats [42]. Together, the results of the present study and our previous work indicate that the behavioral effects of VD_3_ are dependent on used experimental paradigms (stressed or non-stressed). This fact concerning the various effects of VD_3_ on the depression-like state in the long-term OVX rats observed in both our studies might be one explanation for the controversial findings concerning the antidepressant-like effects of VD_3_ in the preclinical and clinical investigations. Further studies are needed to clarify the effects of VD_3_ and the mechanisms of its action in non-stressed and stressed OVX rats of various ages and post-ovariectomy periods. The question of why the behavioral effects of VD_3_ supplementation are completely different in non-stressed and stressed long-term OVX rats is under discussion and requires additional complex investigations.

The antidepressant-like effects of VD_3_ in t long-term OVX rats with CUMS observed in the present study are in agreement with the existing literature. VD_3_ has been found to restore locomotor activity and the anhedonia state in rat/mice models of depression [59,60]. There exist several possible explanations for the antidepressant-like effects of VD_3_ supplementation in adult long-term OVX rats with CUMS. According to the data in the literature, VD_3_ is implicated in the mechanisms of affective-related disorders by the modulation of HPA axis activity and the BDNF pathway [21,22,61]. It is generally accepted that mood disorders are associated with the dysregulation of the HPA axis [40]. Moreover, it has been found that a high level of CS/ACTH promotes neuronal atrophy and decreases the expression of the BDNF protein and mRNA in the hippocampus [39,62]. It has been found that the hyperactivity of the HPA axis produced by chronic stress leads to depressive-like states that can be reversed by antidepressant treatment [39,63]. In the present study, VD_3_ treatment at a dose of 5.0 mg/kg sc resulted in a marked decrease in the serum CS/ACTH levels, similar to the action of the typical antidepressant fluoxetine. Thus, our results indicate that these two pharmacological treatments can restore and normalize the hyperactivity of the HPA axis in long-term OVX rats subjected to CUMS. However, the present results do not allow us to conclude that the mechanism of the antidepressant-like effect of VD_3_ at a dose of 5.0 mg/kg sc is only connected with the normalization of CS/ACTH levels in the blood serum of long-term OVX rats with CUMS, as our findings suggest that the VD_3_ in all investigated doses equally decreased the pathologically elevated CS/ACTH concentrations in the blood of the long-term OVX rats with CUMS.

Another possible explanation for the recovery responses to VD_3_ in the long-term OVX rats with CUMS is related to its effects on the BDNF pathway. BDNF belongs to the neurotrophin family and presents a neurotrophic factor that facilitates neuronal survival and modulates the proliferation and differentiation of neurons in the brain [36]. BDNF has appealed for consideration among candidate downstream effectors implicated in antidepressant effects [29]. Preclinical studies have found that both acute and chronic stress protocols reduce BDNF expression in the hippocampus [64,65]. Depressed patients also revealed a decreased BDNF in their plasma [33,66]. Clinical and animal studies have documented that chronic pharmacotherapy using typical antidepressants (e.g., selective serotonin re-uptake inhibitors) is linked to increased BDNF expression and neurogenesis in the hippocampus [67,68,69,70,71]. Biochemical studies have also found an increased concentration of BDNF after the application of antidepressants [37]. A high number of antidepressants upregulate the expression of BDNF via cyclic AMP-responsive element-binding protein (CREB) signaling [38]. We can assume that VD_3_ is involved in the modulation of BDNF–CREB signaling in long-term OVX rats with CUMS.

Additionally, it has been found that serotonin and BDNF have bidirectional influence—promoting the signaling and gene expression of each other in the hippocampus [72]. Some studies have demonstrated that VD_3_ promotes the secretion of monoamines, including 5-HT [22,73], which may partly explain its beneficial effect on anhedonia and depressive symptoms in long-term OVX rats with CUMS. Further studies should be performed to establish whether the effect of VD_3_ on hippocampal BDNF expression is generated by the serotonergic system or by another neurobiological mechanism.

On the other hand, other neurotrophic factors such as NT-3 and NT-4 are also implicated in the pathophysiology of mood disorders [74,75]. Our study is the first work showing the involvement of the NT-3/NT-4 pathways in the antidepressive-like and anti-anhedonia-like effects of VD_3_ at different doses in adult long-term OVX rats exposed to CUMS. Our results suggest that the normalization of hippocampal concentrations and expressions of neurotrophic factors may be associated with certain aspects of the antidepressant-like effects of VD_3_ in long-term OVX rats with CUMS. Several cross-sectional and cohort studies have reported an inverse association between plasma 25OH-VD concentrations and depression in women [61,76,77,78]. In line with these observations, depressive symptoms occurred more frequently in women with VD deficiency/insufficiency, and VD improved the mood state in such women [79,80,81]. The possible mechanism of VD action might be explained by the stimulation of VDR identified in the different brain structures involved in mood control [13,14]. VD may affect dopaminergic and/or serotoninergic neurotransmitter systems, neurotrophic factors, and/or alter the HPA axis response at the depressive state [20,82,83]. Low VD levels appear in the majority of postmenopausal women [79,80]. Furthermore, estrogen deficiency after menopause decreases the activity of 1α-OHase, which results in the lower synthesis of the functionally active VD forms [84]. Therefore, VD supplementation may be very useful in depressed postmenopausal women with a low level of VD [85]. However, the exact role of VD supplementation in the prevention and treatment of mood disorders associated with menopausal consequences has not been completely established.

In conclusion, this study evidences the anti-anhedonia-like and antidepressant-like effects of the repeated administration of VD_3_ in a chronic stress model of depression and the biochemical and western blotting assays confirm the implications of BDNF/NT-3/NT-4 modulation in the antidepressant-like activity of VD_3_ in long-term OVX adult rats with CUMS. These results determine the direction of further studies to explore the precise mechanism of VD_3_ action due to the necessity of an improvement in the management of depression treatment in females. Multiple neurobiological mechanisms appear to be implicated in mediating the therapeutic effects of VD_3_ in long-term OVX rats with CUMS. The main question of whether VD deficiency results in depression or depression leads to VD deficiency in long-term OVX rats exposed to CUMS is still under discussion.

## 5. Conclusions

This is the first study to show a beneficial effect of VD_3_ supplementation at a dose of 5.0 mg/kg sc on behavioral impairments induced by CUMS in long-term OVX female rats. This work promotes the creation of more effective and novel therapeutic targets and strategies for depression treatment in subjects with long-term estrogen deficiency.

## Figures and Tables

**Figure 1 nutrients-11-01726-f001:**
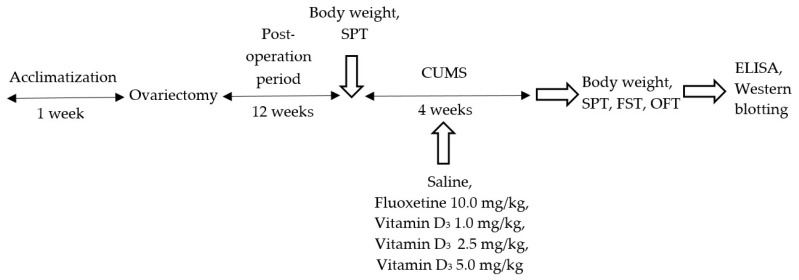
Total scheme of the present experimental study. The antidepressant-like effects of VD_3_ at different doses (1.0, 2.5, and 5.0 mg/kg sc) on the model of depression produced by chronic unpredictable mild stress (CUMS) for 28 days in the long-term (3 months) OVX rats. Sucrose preference (SPT), forced swimming (FST), and open-field (OFT) tests were conducted to examine the depression-like state. Serum corticosterone/ACTH, 25-OH-VD_3_, estradiol levels and hippocampal BDNF, and NT-3/NT-4 expressions were determined by enzyme-linked immunosorbent assay (ELISA) kits and/or western blotting to assess the possible mechanisms of the VD_3_ effects on the depression-like profile in long-term OVX rats subjected to CUMS.

**Figure 2 nutrients-11-01726-f002:**
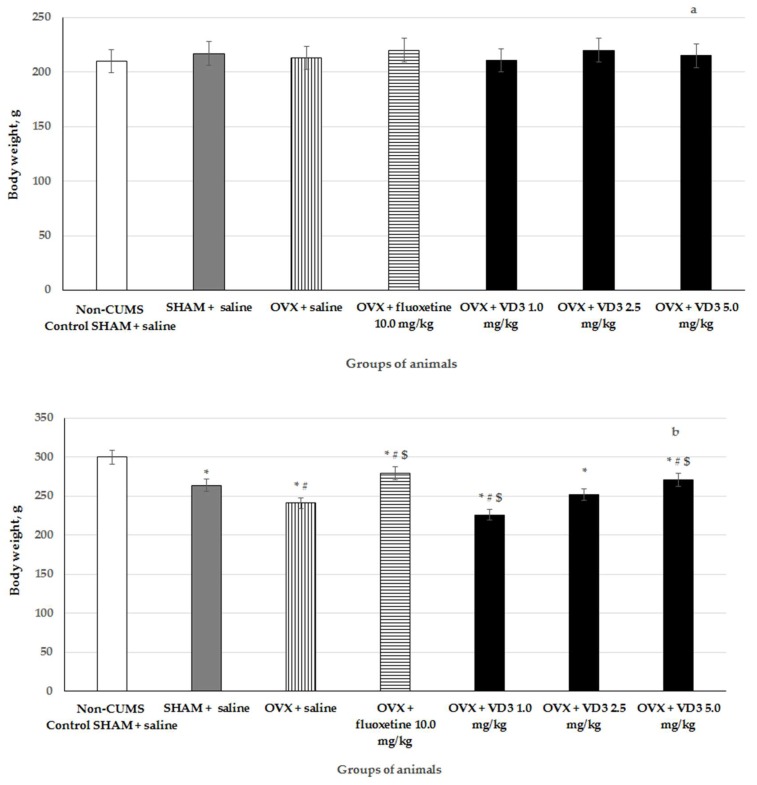
Effects of VD_3_ administered in various doses on the bodyweight of long-term OVX rats subjected to CUMS on day 0 (**a)** the beginning of stress protocol) and on day 28, (**b)** the end of stress protocol). *—*p* < 0.05 versus the non-CUMS SHAM group (control group) treated with saline, #—*p* < 0.05 versus the SHAM group with CUMS treated with saline, $—*p* < 0.05 versus the long-term OVX group with CUMS treated with saline. The data are presented as the mean ± SD; *n* = 7 in each group.

**Figure 3 nutrients-11-01726-f003:**
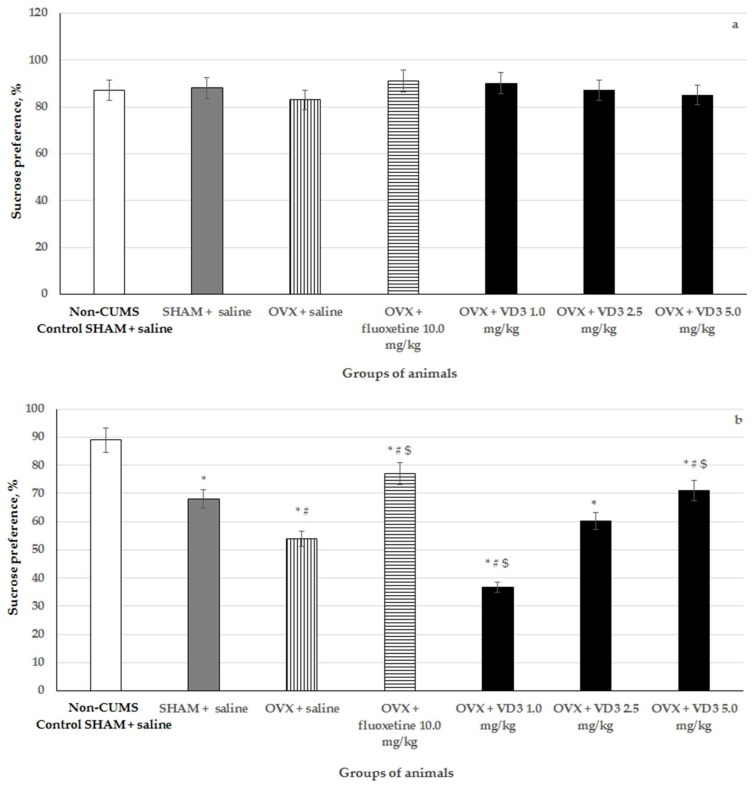
Effects of VD_3_ administered in various doses on sucrose preference in long-term OVX rats before (**a**) and after (**b**) CUMS protocol. *—*p* < 0.05 versus the non-CUMS SHAM group (control group) treated with saline, #—*p* < 0.05 versus the SHAM group with CUMS treated with saline, $—*p* < 0.05 versus the long-term OVX group with CUMS treated with saline. The data are presented as the mean ± SD; *n* = 7 in each group.

**Figure 4 nutrients-11-01726-f004:**
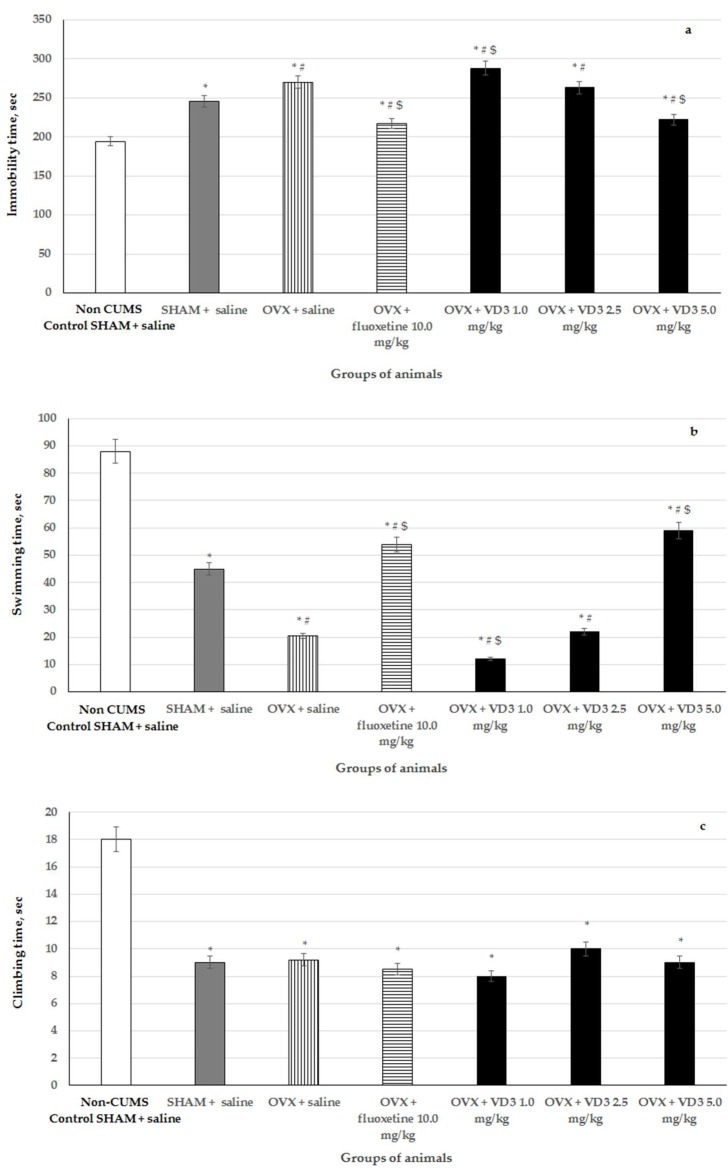
Effects of VD_3_ administered in various doses on depression-like behavior in long-term OVX rats subjected to CUMS in the forced swimming test. (**a**)—immobility time, sec, (**b)**—swimming time, sec, (**c**)—climbing time, sec. *—*p* < 0.05 versus the non-CUMS SHAM group (control group) treated with saline, #—*p* < 0.05 versus the SHAM group with CUMS treated with saline, $—*p* < 0.05 versus the long-term OVX group with CUMS treated with saline. The data are presented as the mean ± SD; *n* = 7 in each group.

**Figure 5 nutrients-11-01726-f005:**
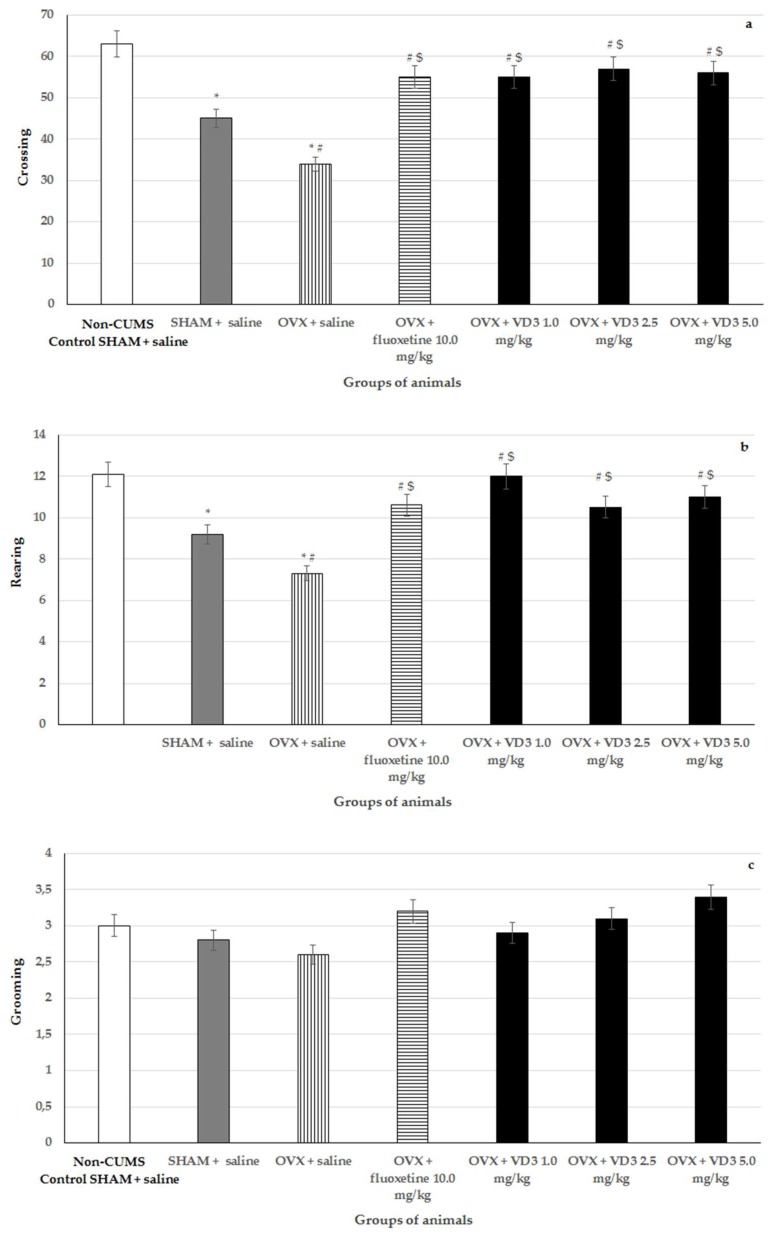
Effects of VD_3_ administered in various doses on the behavior in long-term OVX rats subjected to CUMS in the open field test. (**a**)—crossing, (**b**)—rearing, (**c**)—grooming. *—*p* < 0.05 versus the non-CUMS SHAM group (control group) treated with saline, #—*p* < 0.05 versus the SHAM group with CUMS treated with saline, $—*p* < 0.05 versus the long-term OVX group with CUMS treated with saline. The data are presented as the mean ± SD; *n* = 7 in each group.

**Figure 6 nutrients-11-01726-f006:**
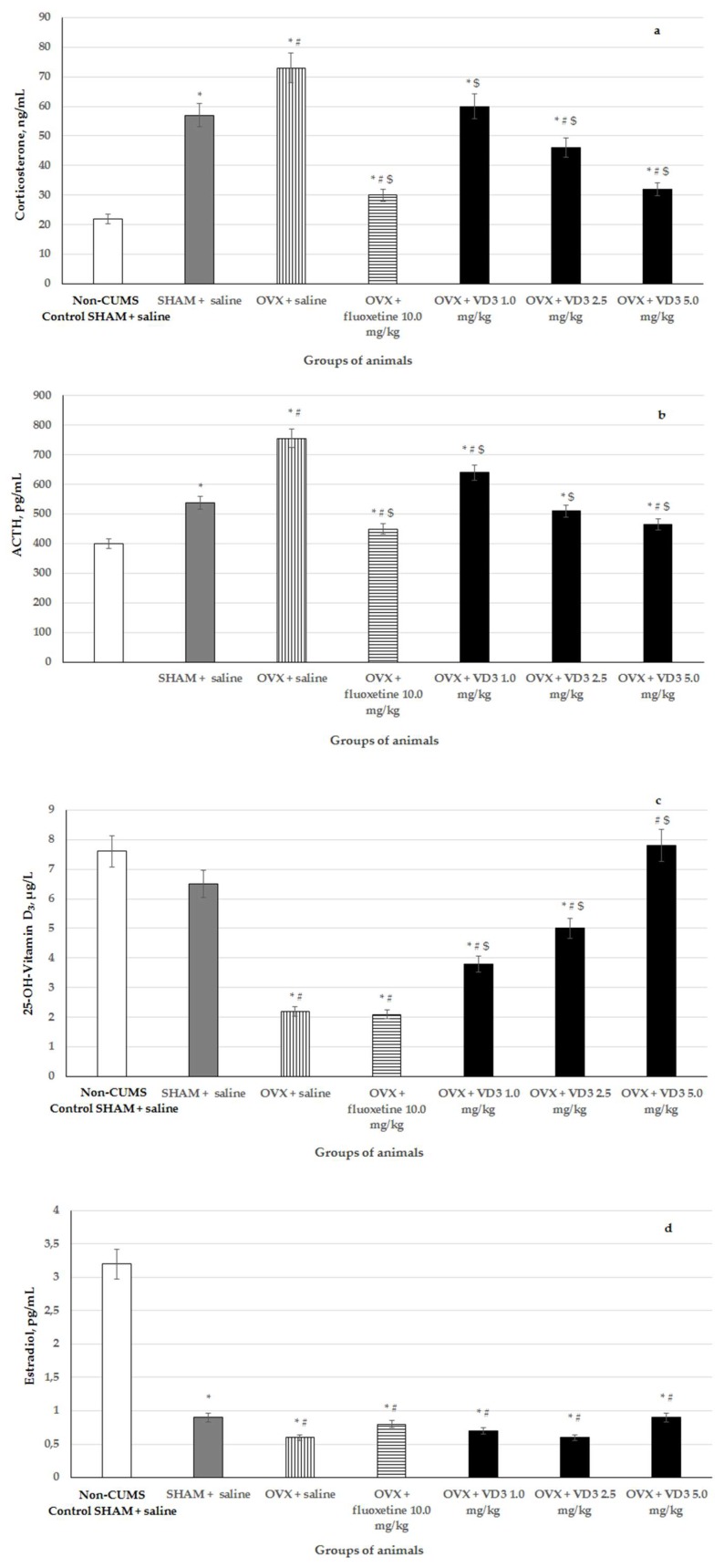
Effects of VD_3_ administered in various doses on serum corticosterone, ACTH, estradiol and 25-OH-VD_3_ levels in the long-term OVX rats subjected to CUMS. (**a**)—Corticosterone, ng/mL, (**b**)—ACTH, pg/mL, (**c**)—25-OH-Vitamin D_3_, µg/mL, (**d**)—Estradiol, pg/mL. *—*p* < 0.05 versus the non-CUMS SHAM group (control group) treated with saline, #—*p* < 0.05 versus the SHAM group with CUMS treated with saline, $—*p* < 0.05 versus the long-term OVX group with CUMS treated with saline. The data are presented as the mean ± SD; *n* = 7 in each group.

**Figure 7 nutrients-11-01726-f007:**
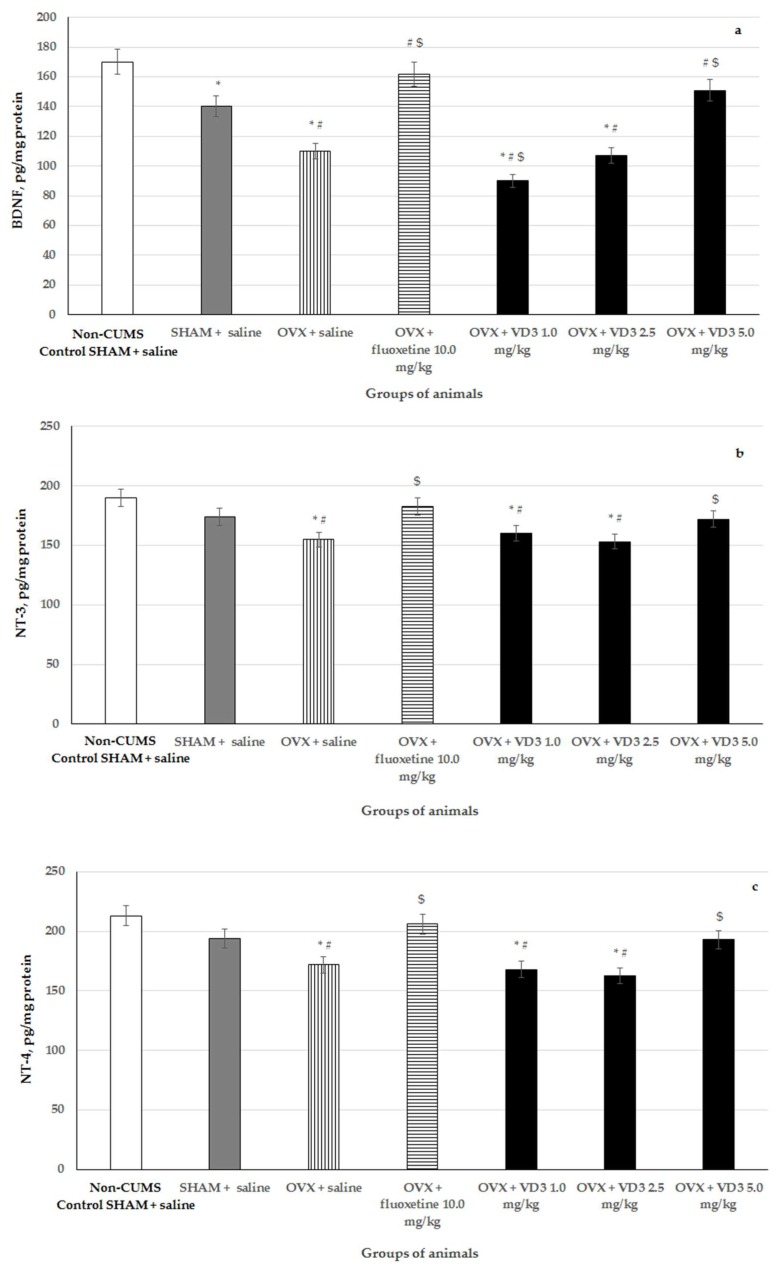
Effects of VD_3_ administered in various doses on hippocampal BDNF (**a**), NT-3 (**b**), and NT-4 (**c**) concentrations in the long-term OVX rats subjected to CUMS. *—*p* < 0.05 versus the non-CUMS SHAM group (control group) treated with saline, #—*p* < 0.05 versus the SHAM group with CUMS treated with saline, $—*p* < 0.05 versus the long-term OVX group with CUMS treated with saline. The data are presented as the mean ± SD; *n* = 7 in each group.

**Figure 8 nutrients-11-01726-f008:**
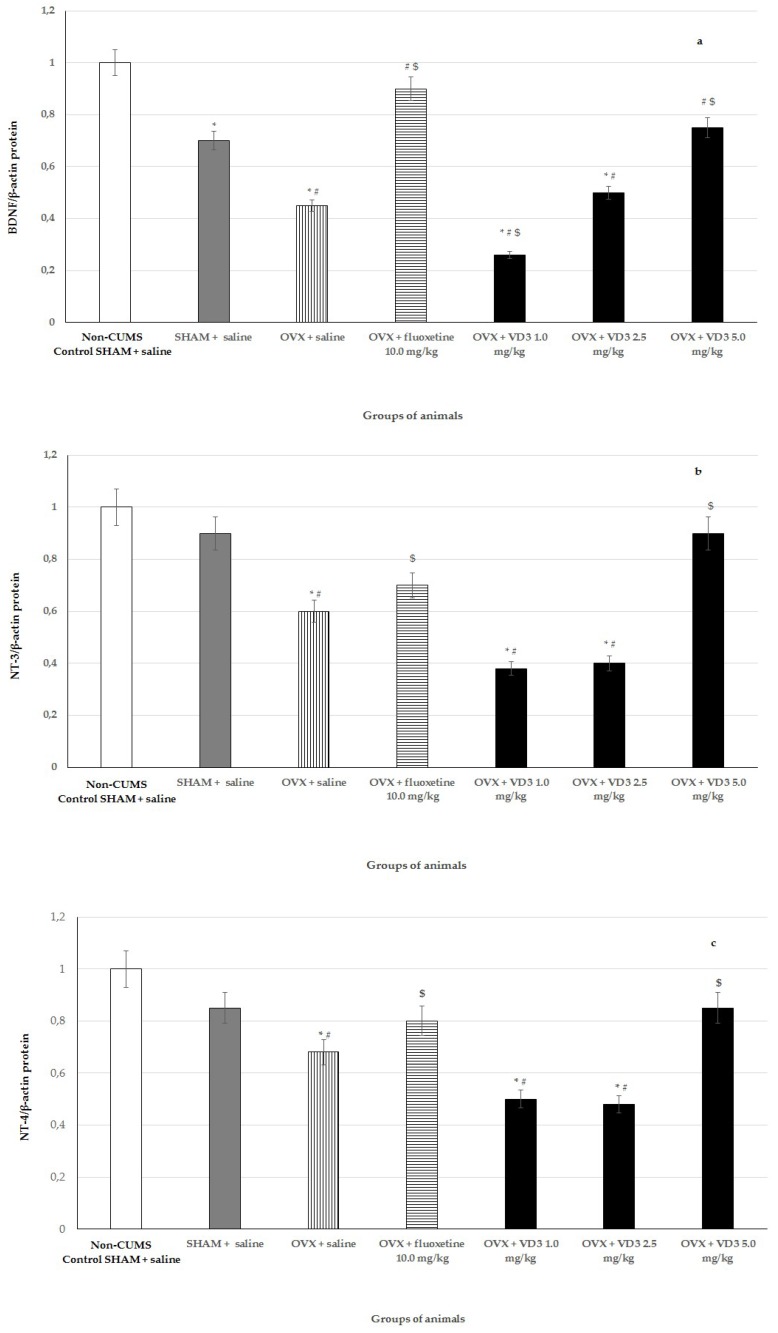
Effects of VD_3_ administered in various doses on hippocampal BDNF **(a)**, NT-3 **(b)**, and NT-4 **(c)** relative expressions in the long-term OVX rats subjected to CUMS. (**d**)1—non-CUMS SHAM (control) rats, 2—SHAM rats with CUMS, 3—OVX rats with CUMS plus saline, 4—OVX rats with CUMS plus fluoxetine (10.0 mg/kg), 5—OVX rats with CUMS plus VD_3_ (1.0 mg/kg), 6—OVX rats with CUMS plus VD_3_ (2.5 mg/kg), 7—OVX rats with CUMS plus VD_3_ (5.0 mg/kg). *—*p* < 0.05 versus the non-CUMS SHAM group (control group) treated with saline, #—*p* < 0.05 versus the SHAM group with CUMS treated with saline, $—*p* < 0.05 versus the long-term OVX group with CUMS treated with saline. The data are presented as the mean ± SD; *n* = 7 in each group.

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
