# Peer review of "Effects of Vitamin D3 in Long-Term Ovariectomized Rats Subjected to Chronic Unpredictable Mild Stress: BDNF, NT-3, and NT-4 Implications"

_nutrients, 2019, doi:10.3390/nu11081726_

Reviewer 1 Report

The manuscript by Koshkina and colleagues aims to explore the relationship between Ovariectomy and Vitamin D3 on depressive-like behaviors in female rats. It appears that the study is well designed but, due to some language and grammar issues, and some errors in the figures, it’s difficult to determine the soundness of the experiments.

One issue is that there is no control groups for the CUMS procedure. Although based on the behavior graphs and body weight table, it would appear that the “sham” group didn’t undergo the CUMS procedure. Is this correct?

Figure 5 appears to be the incorrect figure. It is a repeat of Figure 4 and the figure legend does not match. This is a critical figure for the manuscript and will aid in determination of the other data.

Overall, the authors need to adjust the grammar and sentence structure throughout the manuscript. There are too many examples to list here but the first sentence of the manuscript is an example, “The very specific women state as menopausal period is produced by decrease secretion of the female gonadal hormones by the ovaries”, should be changed to something like, “Menopause is a period in a woman’s life marked by decreased secretion of gonadal hormones”.

Section 2.5 needs more detail. It’s a bit confusing and difficult to determine what the groups are. In the same section, last sentence, it says “Control groups of sham-operated was subcutaneously treated by oil solvent in the volume similarly to those in the OVX rats given with Vitamin D3 or fluoxetine”. Why were these animals administered oil? This is an improper control as the other drugs were dissolved in sterile saline.

Section 2.7. It states that only OVX female rats with CUMS were submitted to the FST. This doesn’t appear to be correct.

Section 2.9 How was the blood drawn? What does “decapitated by similar narcosis for ovariectomy” mean?

Table 1 should be improved. It’s not clear what the table is showing. I think showing “body weight gain” (body weight after CUMS – body weight before CUMS) as a bar graph would be a better way to represent these data.

There are some abbreviations that are never defined such as “VD”, and then other abbreviations that are defined twice, “OVX” for example.

Figure 1 should be improved. Some issues are: Ovariectomy looks like it took place over 12 weeks, but it occurred once and then there was a 12 week “recover” period. There is a red underline in “ovariectomy” should be removed. I believe there should be another “Body Weight” measurement detailed in Figure 1. Blue underline between “Vitamin D3  1.0 mg/kg” should be removed. “ELISA assay” should just be “ELISA”.  

Figure 2 is missing some labels on the x-axis.

Author Response

First of all, we would like to apologize for grammar and sentence structure throughout the whole manuscript. We also very for detailed and very useful scientific analysis of the manuscript. We made total correction of English language using English editing service system of "Nutrients". All corrections  are indicated by red color.

In general, demonstration of different effects of Vitamin D3 in long-term OVX rats on stress model of depression  compared to the non-CUMS SHAM rats and OVX rats with CUMS was the main aim of this study. That is why we did not include all the experimental groups in the previous version of article. Of course, we had SHAM groups with CUMS and without CUMS in the initial experimental protocol. Now, we added all the experimental groups in the corrected version of manuscript.

We corrected all figures and transformed table 1 into figure 1.

Thank you again for your careful review of our article.

Dr. Julia Fedotova

Reviewer 2 Report

In this manuscript, the authors assess the effect of cholecalciferol on the behaviour and neurotrophic singling in an ovariectomized (OVX) rat model after chronic unpredictable mild stress(CUMS). The authors have provided compelling evidence that only a high dose of cholecalciferol injection can rescue CUMS-induced social deficit and downregulation of neurotrophin including BDNF, NT3, and NT4 in OVX rat. This research followed their precious study that cholecalciferol at low dose D rescues OVX induced anxiety-like behaviour. However, this manuscript was not written well, multiple errors in the methods, and also in figures. English needs extensive editing from a native English speaker.

Did the authors examine the levels of 25OHD before and after CUMS in OVX rats? Since OVX reduce the activity of 1,alpha-hydroxylase, the levels of 25OHD could be associated with the behaviour outcomes.

Fig 5, this wrong figure?  The behaviour for ACTH or 25oHD?

The neurotrophin BDNF, NT3 and NT4 were assessed by ELISA and western blot, although these two methods both examine the levels of protein, the results were not consistent,  see NT3 and NT4 levels are different in these two experiments.

In addition, in the western blot results, the bands for NT-3 and NT4 are very similar, also the in NT-4, the last band 6 seems like to be cropped from elsewhere. Did the authors examine the levels of protein in the same gels?  To convince this reviewer, the original gels of all the samples need to be shown.

Figure 6, NT-3 graph, one data set is missing

In the method,  17beta2-E2 treatment was validated, is this 17beta2-E2 in this study? Do OVX rats need 17 beta-E2 replacement?

Table 1 need to be re-organized

Line 97 as well forced swimming… as well as

Author Response

First of all, we would like to apologize for grammar and sentence structure throughout the whole manuscript. We used English editing service system of "Nutrients" to correct English language. All corrections are indicated by red color.

We did not include all the experimental groups in the previous version of manuscript, because of a lot of data. Now, we added all the experimental groups as it was in our initial experimental protocol. We added all original gels.

We corrected all figures.

We did not examine 25-OH-Vitamin D levels before and after CUMS in OVX rats. This criticism can be answered by comparing the different groups of OVX rats with CUMS against the control non-CUMS and CUMS SHAM groups. Thank you very much for this valuable remark for us. We can  perform it in the next study. Moreover, we tested 25-OH-Vitamin D in non-stressed rats of different age in our previous studies (Fedotova et al., 2018).

Thank you again for your careful review of our article.

Dr. Julia Fedotova

Round  2

Reviewer 2 Report

The authors dramatically change the manuscript and also changed the figures. However, many figures did not show the consistency with previous table or figures.

1 Table 1 was changed to figure 2 that one control+saline group was added. However, the value of control+saline in this new figure is the same value as the SHAM+salne, about 300+/-22.3. did the author only replace the value in the new figure?

2 Fig2 in old version, sham rats has 90% sucrose preference, which is changed to 70% in the new figure 3?

3. Fig 3 old vs fig 4 new, the value of sham rats in  are changed to be  the value of control rat in new fig 4, also the cling time of sham rat in old fig 3 (about 17.5sec) is different from that in new figure 4(about 9sec)?

4. Fig 4 old version vs Fig 5 new, same as point 3, why the control in new figure is the same as sham in the old fig?

5.  Regarding the levels of BDNF, NT 3 and NT4, the data of sham animals is now used as control animals, sham animal has a new value?

6. Fig 8, New blot was used to replace the one in last version. In old version, shame control was used as 100%, the other groups were normalised to this control. In the new version, Control + saline was used as 100%, why the relative value of the other groups remained the same as the value in the old version? One could consider that the relative value could change since the control was replaced?

Author Response

Thank you very much for your useful and detailed review of the revised version of our manuscript. I guess there is some confusion and misunderstanding.

As you can see in Section 2.5 of the old version of article we have wrotesham-operated (SHAM) rats without CUMS model treated with solvent...”. In the revised version of article in Section 2.5  we postulated this group as …” SHAM rats without the CUMS model treated with saline (control). In general, …” sham-operated (SHAM) rats without CUMS model treated with solvent…” from old version of article is the same as …” SHAM rats without the CUMS model treated with saline (control)” in revised version of article.

Thus, we did not replace or change values (you can see, for example, in Figure 2 and etc. value sham-operated (SHAM) rats without CUMS model treated with solvent” in old version of article is the same value “300” of  SHAM rats without the CUMS model treated with saline (control) in revised version of article ). This is the same group in old version and revised version of the article, but with different tittle. And as I have already wrote to you previously in my first replay to you, we added only SHAM group with CUMS that was in our initial experimental protocol, but we did not include it in the first initial version of article, only in revised article when you asked about it. We changed only the title of the control group, just add that this is SHAM rats without CUMS plus saline (Control SHAM rats without CUMS).

I believe that such confusion and mix-up was raised, because in the initial version of article we did not primarily postulate …” sham-operated (SHAM) rats without CUMS model treated with solvent...” as control group, however, we did it in revised version of article.

Owing to your detailed review, we deleted this confusion in title for control group, and we indicated this group in all figures as “Non-CUMS Control SHAM + saline” group in the present version of article.

Thank you very much again for your careful review of our article.

Kind regards,

Dr. Julia Fedotova
